# Genomic Insight into Primary Adaptation of *Mycobacterium tuberculosis* to Aroylhydrazones and Nitrofuroylamides In Vitro

**DOI:** 10.3390/antibiotics14030225

**Published:** 2025-02-22

**Authors:** Igor Mokrousov, Violina T. Angelova, Ivaylo Slavchev, Mikhail V. Bezruchko, Simeon Dimitrov, Dmitrii E. Polev, Georgi M. Dobrikov, Violeta Valcheva

**Affiliations:** 1St. Petersburg Pasteur Institute, 197101 St. Petersburg, Russia; felantalion@gmail.com (M.V.B.); brantoza@gmail.com (D.E.P.); 2Faculty of Pharmacy, Medical University, 1000 Sofia, Bulgaria; v.stoyanova@pharmfac.mu-sofia.bg; 3Institute of Organic Chemistry with Centre of Phytochemistry, Bulgarian Academy of Sciences, 1113 Sofia, Bulgaria; ivailo.slavchev@orgchm.bas.bg (I.S.); georgi.dobrikov@orgchm.bas.bg (G.M.D.); 4The Stephan Angeloff Institute of Microbiology, Bulgarian Academy of Sciences, 1113 Sofia, Bulgaria; simo_vets@abv.bg

**Keywords:** *Mycobacterium tuberculosis*, drug resistance, whole-genome sequencing, adaptation

## Abstract

**Background/Objectives:** New anti-tuberculosis compounds are needed to treat patients infected with multi- or extensively drug-resistant *Mycobacterium tuberculosis* strains. Studies based on spontaneous in vitro mutagenesis can provide insights into the possible modes of action and resistance mechanisms of such new compounds. We evaluated the primary response of *M. tuberculosis* in vitro to the action of new aroylhydrazones and nitrofuroylamides. **Methods**: The reference strain H37Rv was cultured on solid media with compounds at increased concentrations relative to MIC. Resistant clones were investigated using whole-genome sequencing and bioinformatics tools to assess the role and potential impact of identified mutations. **Results**: Some of the mutations are significant (based on in silico analysis), located in essential genes, and therefore of particular interest. Frameshift mutations were observed in (i) *Rv2702/ppgK*, which is associated with starvation-induced drug tolerance and persistence in mice, and (ii) *Rv3696c/glpK*, which has been described as a switch on/off mutation associated with drug tolerance. Nonsynonymous substitutions were found in *Rv0506/mmpS2*, which belongs to the Mmp protein family involved in transport and drug efflux, and in *infB*, encoding the translation initiation factor IF-2. **Conclusions**: The primary adaptation of *M. tuberculosis* to the selective pressure of the tested compounds is complex and multifaceted. It involves multiple unrelated genes and pathways linked to non-specific drug tolerance, efflux systems, or mechanisms counteracting oxidative stress.

## 1. Introduction

Despite the existence of effective treatment regimens and prevention measures, the widespread transmission and prevalence of drug-resistant strains of *Mycobacterium tuberculosis*, which are unresponsive to commercially available drugs, remain a significant global health challenge for both developed and developing nations. Multidrug-resistant (MDR) strains pose a growing threat to health security. This issue is of particular concern worldwide, exacerbated by increased human mobility, including illegal and spontaneous migration. Patients infected with drug-resistant strains face limited or no treatment options, often requiring more toxic and costly therapies. The urgent need for novel antimycobacterial agents and innovative development pathways is becoming increasingly evident. The rise and spread of strains resistant to new-generation drugs highlight the necessity for continued exploration of potent compounds with anti-TB activity and the optimization of their chemical structures. Recent studies have summarized the most active synthesized compounds and their progress in preclinical and clinical trials, categorized by chemical structure—amines, aminoalcohols, hydrazides, ureas, thioureas, heterocycles, and others [1,2,3].

When designing new drugs to combat *M. tuberculosis*, it is crucial to consider the bacterium’s unique and chemically resistant cell wall. This structural feature makes anti-TB agents highly specific, as they typically do not affect other pathogenic bacteria, and conversely, most existing antibiotics are ineffective against mycobacteria (with few exceptions). This underscores the need for the development of new, effective anti-TB compounds, along with detailed investigations into their mechanisms of action. Potential candidates for such compounds may be found among previously developed drugs designed for other infections or among new compounds from known classes of antitubercular drugs. In this context, new nitrofuroylamides and aroylhydrazones represent a promising and intriguing area of research.

Nitrofuroylamides have been the focus of research in recent years as potential anti-TB candidates, demonstrating strong antibacterial activity, particularly against mycobacterial species [4,5,6]. Studies suggested that replacing the furan ring with alternative structural motifs frequently leads to a significant reduction or complete loss of antimycobacterial activity [7,8]. Thus, compounds containing a furan moiety could pave the way for discovering potent antitubercular candidates after modifications to the furan core [9,10,11,12,13]. Indeed, the furan heterocycle is incorporated in the chemical structures of numerous drugs from various therapeutic classes, such as 5-nitrofuran derivatives and cefuroxime (antibacterials), darunavir (an HIV protease inhibitor), furosemide (a diuretic), lapatinib (an antineoplastic agent and tyrosine kinase inhibitor), prazosin (an alpha-blocker for hypertension), and ranitidine (a histamine H2 receptor antagonist).

Aroylhydrazones and heteroaroyl hydrazones are another class of chemical compounds that attract attention due to their antibacterial potential. These compounds contain a hydrazone group (–C=NNH–) and exhibit behavior similar to isoniazid but hold significant promise as antimycobacterial agents due to their ability to interact with multiple biological targets within *M. tuberculosis* [14,15,16]. In recent years, many aroylhydrazone and hydrazide–hydrazone derivatives with antimycobacterial activity have been developed as antimycobacterial agents. In most studies, they were three to four times more potent than isoniazid [17,18].

The mechanisms behind the antitubercular activity of nitrofuroylamides and aroylhydrazones remain partly unclear.

Nitrofurans, including nitrofuroylamides, are prodrugs and exert anti-TB activity through the reduction in the nitro group by reductases [4,7,19]. Quantitative structure–activity relationship (QSAR) studies have shown the importance of the nitro group, as nitroaromatic systems significantly increase activity against latent bacteria. The main known mechanism of action of nitrofurans relies on activation by deazaflavin-dependent nitroreductase Ddn, leading to oxidative stress caused by bactericidal reactive oxygen and nitrogen species.

The mechanisms of action of aroylhydrazones include: (a) inhibition of InhA, disrupting mycolic acid synthesis and weakening the bacterial cell wall [20]; (b) additional effects on DNA replication, metal ion chelation, and enzyme inhibition [21], thereby offering a potential strategy to overcome resistance mechanisms associated with isoniazid [22].

Previously, we obtained new nitrofuroylamides and described their properties [6,23]. In particular, based on molecular docking results, compound DO-190 demonstrated favorable binding energies for three out of the four targets examined. Both DO-190 and DO-209 exhibited promising protein–ligand interactions and low MIC values against the reference *M. tuberculosis* strain H37Rv. These findings suggest that both DO-190 and DO-209 possess significant potential, characterized by promising activity and low toxicity, making them strong candidates for further investigation, evaluation, and/or lead optimization.

In our previous studies on hydrazide–hydrazones, the combination of an indole-containing hydrazide–hydrazone fragment with a furan ring led to the creation of a hybrid compound, MLT_FUR, which shows three times better antimycobacterial activity and very low toxicity against HEK-293T human embryonic kidney cell lines [24]. The selectivity index (SI) values and antioxidant activity of the aroylhydrazone derivatives further highlight the high therapeutic potential of the compound, emphasizing its selectivity for mycobacterial cells over human cells. Additionally, the heteroaroyl hydrazone derivative with a thiadiazole fragment, named SNN_VAL, had a low MIC value of 0.0730 µM against the *M. tuberculosis* reference strain H37Rv and low cytotoxicity in the normal human embryonic kidney cell line HEK-293T and mouse fibroblast cell line CCL-1 [25]. The bactericidal activity of the synthesized compounds was demonstrated by molecular docking and experimental studies. The safety profiles and pharmacokinetic properties of these two lead molecules—MLT_FUR and VAL_SNN—are promising, making them strong candidates for further investigation [26].

It should be noted that, similar to the situation with first- and second-line drugs, clinical *M. tuberculosis* strains have demonstrated their capacity to rapidly acquire resistance to new drugs. This was shown by population-based studies as well as by the analysis of clinical isolates recovered from patients during long-term treatment [27,28,29]. Therefore, it is important to stay one step ahead of the microbe and, already at the initial stages of drug design and development, gain insight into potential mechanisms of *M. tuberculosis* resistance to the new compounds. In this sense, the development of new candidate anti-TB compounds should be accompanied by studies on *M. tuberculosis* resistance mechanisms to these compounds. Studies based on spontaneous in vitro mutagenesis can provide insight into understanding possible resistance mechanisms [30,31,32]. Cultivation of bacteria on a medium with increased concentrations of a compound can select for mutations associated with the primary response of bacteria. Previously, we demonstrated the emergence of multiple spontaneous resistant mutants selected under conditions of elevated concentrations of nitrofuran (4-(3-hydroxyphenyl)piperazin-1-yl)(5-nitrofuran-2-yl)methanone, which involved different and unrelated genes [6].

In this study, we evaluated the response of *M. tuberculosis* to the aforementioned aroylhydrazones (MLT_FUR and VAL_SNN) and nitrofuroylamides (DO-190 and DO-209) using the in vitro mutagenesis approach followed by genomic analysis of the spontaneous resistant variants. These compounds are of particular interest because they have already demonstrated potent in vitro antimycobacterial activity and promising pharmacological properties [6,23,24,25,26]. Thus, their further investigation as prospective antitubercular drugs is well-founded.

## 2. Results and Discussion

Genomic and bioinformatics analysis, along with a comparison to data from the wild-type parental strain, identified resistant *M. tuberculosis* clones with mutations that emerged during culturing on media containing elevated concentrations of MLT_FUR, VAL_SNN, and DO-209 (Appendix A). These mutations included single-nucleotide polymorphisms (SNPs) or long/short insertions/deletions, located in coding sequences or intergenic regions. The significance of the nonsynonymous mutations—i.e., their potential role in altering protein function—was assessed in silico using SIFT- and PAM1-based methods. Information on the mutant genes was retrieved from the Mycobrowser resource and published articles.

Results for individual samples with mutations are presented in Appendix A, while information on the identified mutations and mutated genes is summarized in Table 1 and Table 2 and discussed in detail below.

### 2.1. Aroylhydrazones Resistance

A nonsynonymous mutation was detected in the *Rv3755c* gene at position 302 A>G, affecting codon 101. *Rv3755c* encodes a hypothetical conserved protein with an unknown function, according to the Mycobrowser resource. At the same time, the *Rv3755c* gene network includes several ABC transporter genes, and on the chromosome, *Rv3755c* is located near these genes and is predicted to be co-expressed with them (Appendix A). The ABC transporter superfamily includes proteins involved in the transport of inorganic ions, polysaccharides, amino acids, and peptides. ABC transporter proteins are also known to play a role in the efflux of antibiotics from the cell [33]. The mutation in *Rv3755c* may be causatively related to the nonspecific adaptation of bacteria to the selective pressure of the MLT_FUR compound, potentially by increasing its active efflux from the cell. In three Himar1 transposon mutagenesis studies, *Rv3755c* was identified as a nonessential gene for H37Rv growth in vitro [34,35,36]. However, it is noteworthy that a slow-growing H37Rv *Rv3755c* mutant was obtained using Himar1-based transposon mutagenesis [37]. One of the key responses of *M. tuberculosis* to drug pressure is a decrease in growth rate, which correlates with drug tolerance [38]. This suggests that drug tolerance may be one of the primary responses of *M. tuberculosis* to this compound. In turn, we cautiously hypothesize that *Rv3755c*, currently defined as a protein with unknown function, may be related to the ABC transporter efflux system.

Two mutations were detected in one MLT_FUR-resistant clone: a frameshift mutation in *Rv2702 (ppgK)* and a nonsynonymous mutation in *Rv0506 (mmpS2)*.

The frameshift in *Rv2702 (ppgK)* affected codon 184, while the total gene length is 265 amino acids. We assume this mutation led to gene inactivation and was significant, as the resulting protein would be truncated by 30%. *ppgK* is defined as an essential gene by Himar1 transposon mutagenesis in H37Rv [39] and is required for growth in the C57BL/6J mouse spleen [40]. This gene is likely related to starvation-induced drug tolerance and persistence of *M. tuberculosis* in mice [40,41]. Under nutrient-limited conditions, the production of the outer membrane lipid phthiocerol dimycocerosate (PDIM) is required for *M. tuberculosis* drug tolerance. However, PDIM production was prevented in a *ppgK* SNP mutant [41]. The gene–gene network of *ppgK* (Appendix A) includes *sigA*, which encodes the RNA polymerase sigma factor SigA, an important transcription initiation factor. Wu et al. [42] suggested that *sigA* modulates the expression of genes that contribute to virulence, enhancing growth in human macrophages and during the early phases of pulmonary infection in vivo. This effect may be partly mediated by increased resistance to reactive oxygen intermediates. *sigA* and *ppgK* are linked not only by their chromosomal proximity but also by co-expression (score 0.044) (Appendix A). The almost complete inactivation of polyphosphate glucokinase PpgK (this deletion was found in 94% of sequencing reads in this clone) could be beneficial for mycobacterial survival, as it may hypothetically activate a more general stress response, indirectly leading to tolerance to the compound.

A nonsynonymous substitution was detected in *Rv0506*, which codes for the conserved membrane protein MmpS2. This mutation appears to be non-significant, as assessed by PAM1 and SIFT. However, Mmp proteins are involved in transport, and some of them mediate drug tolerance through efflux pump mechanisms [43]. The gene network of *mmpS2* (Appendix A) shows that it is linked not only by gene neighborhood but also by co-expression to *hemC*, *hemA* (essential genes for intermediary metabolism and respiration), and *mmpL2*. The latter is an essential gene for in vitro growth of H37Rv [36], codes for a conserved transmembrane transport protein, and belongs to the efflux pump family [44]. In this context, it is tempting to speculate that *mmpS2* may be involved in efflux-mediated drug tolerance, and the observed mutation may be beneficial for *M. tuberculosis* growth under elevated concentrations of MLT_FUR.

A frameshift insertion mutation in *Rv3696c/glpK* involved a change within the homopolymeric tract ACCCCCC>ACCCCCCC, leading to the inactivation of *glpK*. This mutation affects codon 191; since the gene length is 517 amino acids, the resulting protein is truncated by 60% and rendered nonfunctional. GlpK is a key enzyme involved in the regulation of glycerol uptake and metabolism and has been demonstrated to be essential for the in vitro growth of H37Rv [36]. This specific mutation has been described as conferring transient drug tolerance to different drugs through phase variation [31,38,45]. Phase variation, an adaptive mechanism, is based on reversible on/off gene switching, particularly through insertions or deletions within homopolymeric tracts such as CCCCCCCC. These mutations result from strand slippage mispairing during replication and occur at a particularly high rate in bacteria with deficient DNA mismatch repair systems. *M. tuberculosis* is an example of such a species. Inactivating frameshift C insertions in *glpK* produce small colonies that exhibit heritable increases in MIC to different drugs [38]. Along with metabolic shifting and activation of efflux pumps, a reduced growth rate is among the key responses of *M. tuberculosis* to drug pressure. Thus, a clone bearing this mutation in *glpK* (in 83% of sequencing reads) was likely rapidly selected in response to the action of this compound.

While all the above mutations were nonsynonymous or frameshift, a mutation detected in *Rv3366/spoU* was silent. *spoU* encodes tRNA/rRNA methylase SpoU, which is involved in methylation and belongs to the Information Pathways gene category. Most likely, this synonymous mutation did not influence the bacterial response to the compound’s action, although the fact that it occurred in a methylase gene may provide some insight, as DNA methylation can modulate gene expression [46]. Synonymous mutations are known to affect gene expression by altering mRNA stability, mRNA secondary structure, splicing processes, and translational kinetics [47,48,49]. Accordingly, this synonymous mutation in the *spoU* gene may not have been randomly selected but could be related to the survival of this clone under elevated concentrations of the aroylhydrazone VAL_SNN.

In summary, the spontaneous mutant clones resistant to aroylhydrazones MLT_FUR or VAL_SNN harbored mutations in the following genes: *ppgK*, *mmpS2*, *glpK*, *Rv3755c*, or *spoU*. Gene–gene network analysis revealed that these five genes are not connected (Figure 1A), meaning they are not linked to each other and are involved in different pathways. Furthermore, when *inhA*, which codes for the target of aroylhydrazones, was added to these genes, the network did not establish any connections between them (Figure 1B).

### 2.2. Nitrofuroylamides Resistance

Two mutations in a DO-209-resistant clone were identified in *Rv2839c/infB* and *Rv3053c/nrdH*, both of which belong to the Information Pathways gene category.

A nonsynonymous mutation in *Rv2839c/infB* was assessed in silico to be significant by both PAM1 and SIFT analyses. The gene *Rv2839c* codes for the probable translation initiation factor IF-2 (InfB), one of the main components required for translation initiation. It is an essential gene, as consistently demonstrated by various transposon mutagenesis studies [34,35,36,37]. It is located at the core of the translation machinery, and its gene network includes various ribosomal proteins such as RpLD, RpsB, RpsK, RpsG, RplC, RplB, RpsE, and RbfA (Appendix A). The role of translation-associated factors may be complex [50,51,52,53]. For example, in *M. smegmatis*, disruption of *lepA*, which codes for a translation-associated elongation factor, increased rifampin tolerance [50]. For another initiation factor, IF3, it was demonstrated that its interaction with the ribosome assembly factor RbfA contributes to increased stress tolerance in *E. coli* [52]. Speculatively, this mutation in *infB* in the *M. tuberculosis* clone resistant to nitrofuroylamide DO-209 could indeed be linked to tolerance to this compound, although the mechanism remains unknown.

Another mutation in the same clone was a large 14-base-pair deletion in the intergenic region upstream of *Rv3053c/nrdH*, located at position -223 relative to the start codon. This gene is essential for bacterial growth [35,36] and codes for the glutaredoxin electron transport protein NrdH, which is involved in the electron transfer system for the ribonucleotide reductase system [54]. It should be noted that redox sensing and regulation processes in *M. tuberculosis* are critical for its adaptation to the host’s hostile environment, including oxidative stress caused by reactive oxygen species (ROS) and reactive nitrogen species (RNS). *M. tuberculosis* possesses a complex redox network that depends on small molecules such as NADPH and mycothiol, as well as proteins that facilitate electron transfer through cysteines in a CXXC motif [55]. In particular, *M. tuberculosis* redox homeostasis is maintained through the thioredoxin system, which plays a major role in survival under redox stress. NrdH, a thioredoxin-like protein, is one of its components [56]. On the chromosome, *nrdH* is the first gene in the array of *nrd* genes and is linked to other *nrd* genes in the gene–gene network (Appendix A). Given the role of NrdH in counteracting oxidative stress, it is noteworthy that nitrofurans are among the compounds that can induce such stress in *M. tuberculosis*. The Ddn nitroreductase targets the nitro moiety in nitroimidazole drugs, leading to the production of bactericidal nitric oxide, a reactive nitrogen species (RNS) [8]. In this context, the described mutation in the intergenic region, even though distantly located from the start codon of *nrdH*, may have some impact on countering oxidative stress caused by the action of nitrofuroylamide.

We further performed gene–gene interaction analysis on genes with mutations in nitrofuran-resistant spontaneous mutants. In addition to the two genes mentioned above, we included six other genes previously reported to harbor mutations in resistant clones [10]. Our earlier study of spontaneous mutants resistant to the nitrofuran DO-166 identified six genes with mutations in resistant clones: *Rv0224c*, *fbiC*, *iniA*, *Rv1592c*, *Rv1580c*, and *Rv1639c* [10]. Some of these genes are known to be involved in drug tolerance, such as *iniA* and the lipase/esterase *Rv1592c* [57,58], or in counteracting oxidative and nitrosative stress, such as *Rv0224c* and *fbiC* ([10] and references therein).

We performed gene–gene interaction analysis on these eight genes with mutations in nitrofuran-resistant spontaneous mutants. However, no links between these genes were found, indicating that they represent different and independent metabolic pathways and adaptation strategies (Figure 2A). When *ddn*, which codes for the nitrofuran-activating enzyme, was added to these genes, most of the proteins/genes remained unlinked, with only a single edge connecting *ddn* and *fbiC* (Figure 2B). Indeed, *fbiC* belongs to the F420 biosynthesis pathway, which includes the nitroreductase Ddn.

## 3. Materials and Methods

### 3.1. Chemistry Experimental Procedures

The compounds used in this study were aroylhydrazones (MLT_FUR, VAL_SNN) and nitrofuroylamides (DO-190, DO-209) (Table 3). Their synthesis and characteristics were described in detail in our previous studies [6,23,24,25,26].

### 3.2. Determination of the Minimal Inhibitory Concentration (MIC)

The MIC for the *M. tuberculosis* reference strain H37Rv was determined by the resazurin microtiter plate assay (REMA) as described previously [6,59]. A 3-week culture of *M. tuberculosis* was transferred into a sterile, dry tube containing glass beads. The tube was vortexed for 30–40 s, followed by the addition of 5 mL of Middlebrook 7H9 Broth (Becton Dickinson, Franklin Lakes, NJ, USA). The bacterial suspension’s turbidity was adjusted to 1.0 McFarland units (approximately 3 × 10^8^ bacteria/mL) and then diluted 20-fold using Middlebrook 7H9-10% OADC medium. The same medium was used to prepare a 1:100 *M. tuberculosis* control (1% population). Stock solutions of the compounds in DMSO (10 mg/mL) were diluted with Middlebrook 7H9-10% OADC medium to a concentration of 800 µg/mL. A 200 µL aliquot of this solution was added to the second row of a 96-well microtiter plate, where 2-fold serial dilutions were performed. Row 10 served as the *M. tuberculosis* suspension control, row 11 as the 1% control (10-fold diluted culture), and row 12 as the blank control for optical density measurements (200 µL of growth medium). A 100 µL bacterial suspension was added to each well, except for rows 11 (1% control) and 12 (blank medium), bringing the total volume in each well to 200 µL. The plates were incubated at 35 °C for 7 days. After incubation, 30 µL of a 0.01% aqueous resazurin solution (Sigma, St. Louis, MO, USA) was added to each well, and incubation continued for an additional 18 h at 35 °C. Fluorescence was measured using a FLUOstar Optima plate reader (Legacy Instruments, New York, NY, USA). Bacterial viability was assessed by comparing the mean fluorescence values (±CI at α = 0.05) in the control wells (row 12, blank; row 11, 1% control) with those in the wells containing the test compounds. The minimum inhibitory concentration (MIC) was defined as the lowest compound concentration at which fluorescence either plateaued or was statistically similar (*t*-test) to that of the 1% control.

The REMA MIC determination was also conducted for Isoniazid, a known antibiotic (MIC 0.062 µg/mL), to validate the experimental conditions used for MIC determination of the tested aroylhydrazones and nitrofuroylamides.

### 3.3. Spontaneous Mutant Selection, WGS, and Bioinformatics

A suspension of the reference strain H37Rv (ATCC 25618, Manassas, VA, USA) was prepared at 10^8^ CFU/mL, and 1 mL was applied to a solid Lowenstein–Jensen culture medium on a Petri dish with an increased concentration of a compound (two-fold to ten-fold MIC previously determined, see Table 3); a detailed laboratory protocol is available on request from corresponding authors.

The Petri dishes were incubated at 37 °C for 4 weeks, after which they were visually inspected, and the grown colonies were randomly selected for further molecular analysis. Since the clones were grown on elevated concentrations of the tested compounds (relative to the MIC of the parental strain), we operationally defined them as “resistant”.

Bacterial DNA was extracted using a cetyl trimethyl ammonium bromide-based method and used for whole-genome sequencing (WGS). DNA fragmentation was performed using the FTP method [60], followed by the ligation of universal adaptors with T4 DNA ligase. DNA libraries were amplified using barcoded primers and sequenced on the DNBSEQ-G50 platform (MGI, Shenzhen, China) via paired-end sequencing. Alternatively, DNA libraries were prepared using the Illumina DNA Prep kit (San Diego, CA, USA) and sequenced on the NextSeq 500 system. The DNA of the wild-type strain H37Rv cultured on the medium without compounds was also subjected to WGS. The WGS data were deposited in the NCBI Sequence Read Archive (SRX27529710, SRX27350881, SRX27350880, SRX27350879, SRX27350878, SRX27350877, SRX27350874, SRX26650431, SRX26650430, SRX26650429).

Genomic data were processed using bioinformatics programs. WGS fastq files were mapped to the reference strain H37Rv NC_000962.3 genome using the SAM-TB online tool and the results were checked by Geneious R package (Biomatters, Auckland, New Zealand).

The gene–gene interaction network was built using STRING (https://string-db.org/cgi/about.3 (accessed on 30 January 2025)). Amino acid substitutions were assessed using PAM1 (https://en.wikipedia.org/wiki/Point_accepted_mutation (accessed on 30 January 2025)) and the SIFT (Sorting Intolerant From Tolerant) online tool (https://sift.bii.a-star.edu.sg/www/SIFT_seq_submit2.html (accessed on 30 January 2025)) based on the analysis of the UniProtKB/Swiss-Prot and TrEMBL databases.

Information on *M. tuberculosis* genes was taken from https://mycobrowser.epfl.ch/ (accessed on 30 January 2025) and based on a search in PubMed using “tuberculosis” and gene names as keywords.

## 4. Conclusions

To conclude, the primary adaptation of *M. tuberculosis* to the selective pressure of the studied compounds is complex and multifaceted. It involves multiple unrelated genes and pathways, linked to nonspecific tolerance mechanisms such as drug efflux systems (for aroylhydrazones and nitrofuroylamides) or mutations selected in response to oxidative stress (for nitrofuroylamides).

Essentially, the tolerance-conferring mutations reflect either changes in a drug target or alterations in bacterial physiology that enable the bacterium to survive the drug’s effects. However, our study demonstrates that short-term in vitro exposure of the slow-growing *M. tuberculosis* mostly promotes the emergence of mutations that counteract stress conditions.

This study revealed that different mutations were detected in some clones resistant to the tested aroylhydrazones and nitrofuroylamides (i.e., clones grown on elevated concentrations of the tested compounds, relative to their MIC). Some of these changes were assessed in silico to be significant and involved essential genes, making them of particular interest. Frameshift mutations occurred in the *Rv2702* gene, which codes for the PpgK enzyme, a protein related to starvation-induced drug tolerance and persistence in mice. A deletion within the homopolymeric CCCCCCCC tract in *Rv3696c/glpK* was previously shown to be linked to multidrug tolerance. Nonsynonymous substitutions were found in *mmpS2*, which belongs to the Mmp family of proteins involved in transport and drug efflux, and in *infB*, which codes for the important translation initiation factor IF-2.

We hope that our results will be useful for further follow-up studies aimed at advancing these antibacterials and elucidating their modes of action.

This study focused on the well-characterized laboratory strain H37Rv to gain primary and preliminary insights into the mechanisms of mycobacterial resistance to the tested compounds as potential anti-TB drugs. However, *M. tuberculosis* is a genetically heterogeneous species, and some of its lineages, genotypes, and certain epidemic clones are of particular clinical or epidemiological significance. In this context, further research should focus on an expanded collection of MDR and susceptible clinical isolates from high-burden regions to investigate how different lineages within *M. tuberculosis* can (or cannot) adapt to the selective pressure of these compounds.

## Figures and Tables

**Figure 1 antibiotics-14-00225-f001:**
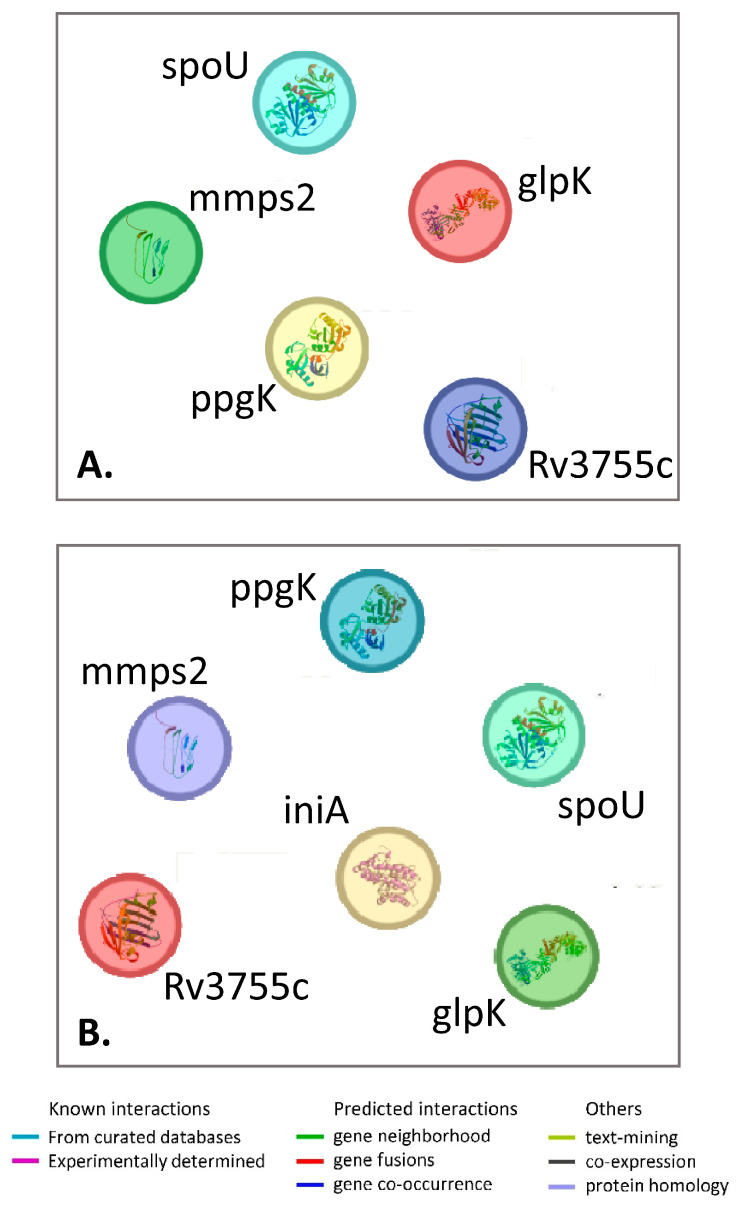
Gene–gene network of genes mutated in aroylhydrazone-resistant clones in vitro. (**A**) Network of five genes detected in this study. (**B**) The same network with added *inhA*, coding for a target of aroylhydrazones InhA. The network was built using STRING (https://string-db.org/cgi/about.3 (accessed on 30 January 2025)). Note the lack of connecting links between the genes, which reflects that the genes are not interacting, to the best of current knowledge.

**Figure 2 antibiotics-14-00225-f002:**
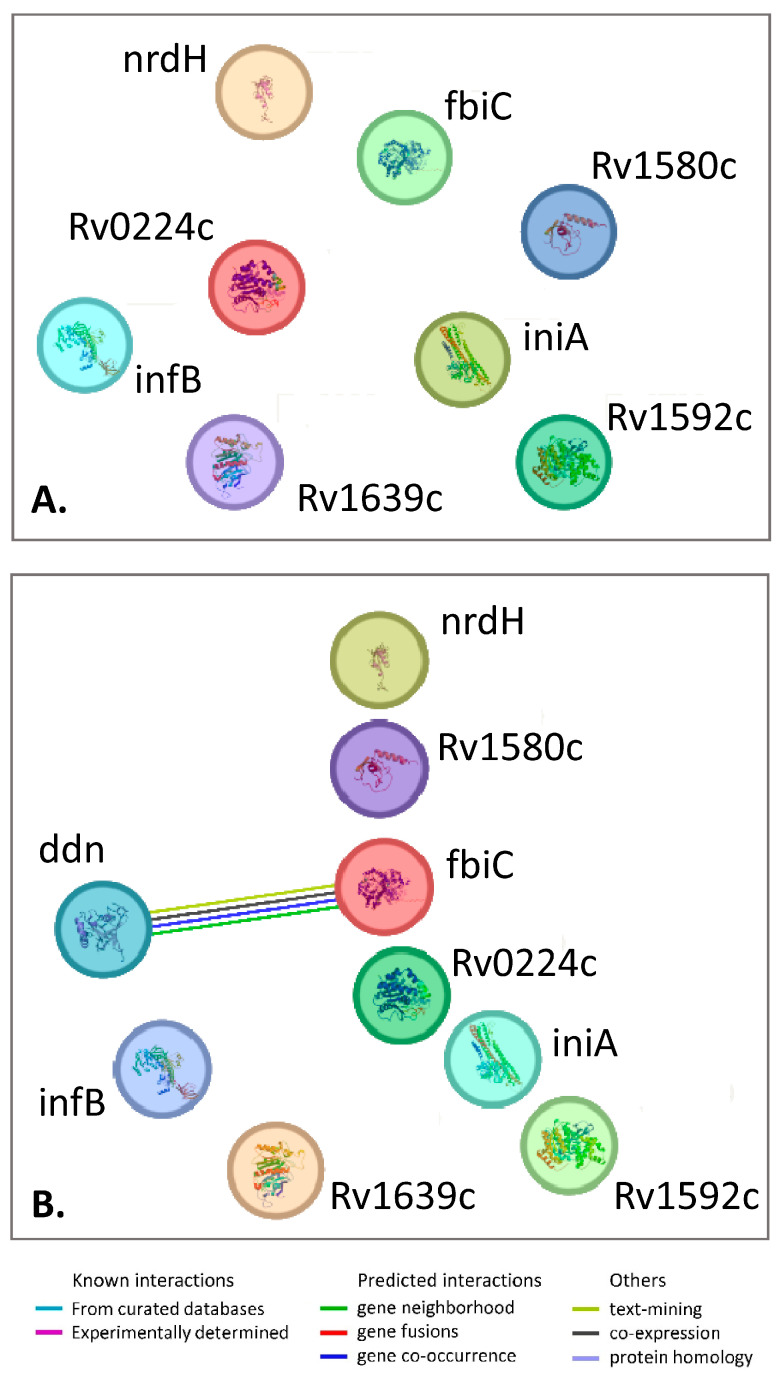
Gene–gene network of genes mutated in nitrofurane-resistant clones in vitro. (**A**) Network of eight genes, including two genes in this study and six previously reported genes [10]. (**B**) The same network with added *ddn* coding for the nitrofurane-activating enzyme Ddn. The network was built using STRING (https://string-db.org/cgi/about.3 (accessed on 30 January 2025)). Note the lack of connecting links between the genes for most of the gene pairs, which reflects that the genes are not interacting, to the best of current knowledge.

**Table 1 antibiotics-14-00225-t001:** Summary of likely significant mutations found in spontaneous mutants of *M. tuberculosis* strain H37Rv grown on solid media with increased concentrations of the tested compounds.

Compound	Position and Mutation in Genome	Gene Label and Name	Position and Mutation in Gene	Codon, Nucleotide, and Amino Acid Change	PAM1	SIFT P	Type of Mutation
MLT_FUR	4201587 T>C	*Rv3755c*	302 A>G	101 CAC>CGC, H>R	10	>0.05	Nonsynonymous
MLT_FUR	3017408 del_C	*Rv2702/Ppgk*	551 del_C	184	-	-	Frameshift (codon 184; gene length 265 aa)
MLT_FUR	597110 A>G	*Rv0506/mmpS2*	352 A>G	118 AGC>GGC, S>G	21	>0.05	Nonsynonymous
VAL_SNN	3777769 C>T	*Rv3366/spoU*	33 C>T	11 ATC>ATT, I>I	-	-	Synonymous
VAL_SNN	4139183 A>AC (ACCCCCC> ACCCCCCC)	*Rv3696c/glpK*	573 T>GT	191	-	-	Frameshift (codon 191; gene length 517 aa)
DO-209	3147075 A>C	*Rv2839c/infB*	799 T>G	267 TTC>GTC, F>V	1	0.00	Nonsynonymous
DO-209	3415181 del_CACCTAGGGGGTGG	*Rv3053c/nrdH*	-223	-	-	-	Intergenic region, deletion

**Table 2 antibiotics-14-00225-t002:** Information on genes with mutations identified in resistant clones derived from *M. tuberculosis* strain H37Rv grown on solid media with increased concentrations of the tested compounds.

Gene Label and Name	Protein Name	Protein Function	Gene Category
*Rv3755c*	Rv3755c	Hypothetical protein	Conserved hypotheticals
*Rv2702/Ppgk*	Polyphosphate glucokinase Ppgk	Catalyzes the phosphorylation of glucose using polyphosphate or ATP as the phosphoryl donor.	Intermediary metabolism and respiration
*Rv0506/mmpS2*	Membrane protein MmpS2	Unknown	Cell wall and cell processes
*Rv3366/spoU*	tRNA/rRNA methylase SpoU	Causes methylation	Information pathways
*Rv3696c/glpK*	Glycerol kinase GlpK	Acts in the rate-limiting step in glycerol utilization. Key enzyme in the regulation of glycerol uptake and metabolism	Intermediary metabolism and respiration
*Rv2839c/infB*	Probable translation initiation factor IF-2 InfB	IF-2, one of the essential components for the initiation of protein synthesis	Information pathways
*Rv3053c/nrdH*	Glutaredoxin electron transport protein NrdH	Involved in the electron transfer system for the ribonucleotide reductase system	Information pathways

**Table 3 antibiotics-14-00225-t003:** Aroylhydrazones and nitrofuroylamides used in this study.

Designation	Name	Structure	MIC * µg/mL	Reference
MLT_FUR	*N*′-[(*E*)-(5-methoxy-1*H*-indol-3-yl)methylidene]furan-2-carbohydrazide	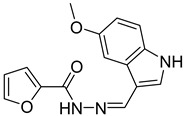	0.07	[24,26]
VAL_SNN	*N*′-[(*E*)-(3-hydroxy-4-methoxyphenyl)methylidene]-4-methyl-1,2,3-thiadiazole-5-carbohydrazide	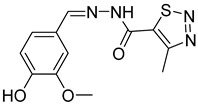	0.25	[25]
DO-190	5-nitrofuran-2-yl)(4-(pyridin-2-yl)piperazin-1-yl)methanone	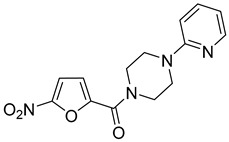	0.06	[6,23]
DO-209	(5-nitrofuran-2-yl)(4-(4-nitrophenyl)piperazin-1-yl)methanone	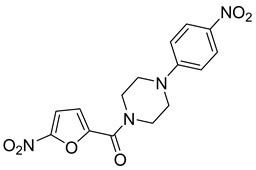	0.02	[6,23]

* MIC, minimal inhibitory concentration. MIC values are given for the *M. tuberculosis* reference strain H37Rv.

## Data Availability

Data are contained within the article and Appendix A. The WGS data (fastq files) were deposited in the NCBI Sequence Read Archive (SRX27529710, SRX27350881, SRX27350880, SRX27350879, SRX27350878, SRX27350877, SRX27350874, SRX26650431, SRX26650430, SRX26650429).

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
