# Peer review of "Genomic Insight into Primary Adaptation of Mycobacterium tuberculosis to Aroylhydrazones and Nitrofuroylamides In Vitro"

_antibiotics, 2025, doi:10.3390/antibiotics14030225_

Round 1
Reviewer 1 Report
Comments and Suggestions for Authors
Please accept my congrtulations on this beautiful piece of work. Broadly I think the study is very useful in helping us understand mechanism of drug resiatance and action in M.tb.
I have a few minor comments:
Line 16 in-vitro hyphenate
Line 21, in silico significant, I m not sure what that means, please rephrase
Line 34-37, this is a very long sentence, plz shorten
The entire introduction seems to be rewritten, it’s a haphazard mix as of now. The authors should t try to write a coherent story which in my view can start by saying TB needs new modalities, a few of these are … these act by … and why they chose the ones they did. This may be followed by a few lines on their methodology as to why they decided on this particular approach. This is only a suggestion alternative story lines that make the paper a worthy read may also be explored.
Line 267, regarding NrdH this protein has been biochemically and structurally characterised Phulera, Mande, 2013 Biochemistry.
soon after Line 267 the authors present a very useful discussion on redox related proteins, I would like to bring to the attention of authors that Mtb has a very interesting redox network relying on either small molecules like NADPH, mycothiol or proteins that help transfer electrons by utilizing Cysteines in a CXXC motif, for eg see Phulera et al Journal of the Indian Institute of Science 2014.
Figure 2 needs to be improved, bigger labels can be a starter, also please have it looked at some other colleagues and see if any other suggestions can make it more useful
The conclusions can really use some more work, please take a moment here to review in a little more details here what some implications of your results are
Comments on the Quality of English Languageneeds work
Author Response
Answers to Reviewer 1
Please accept my congratulations on this beautiful piece of work. Broadly I think the study is very useful in helping us understand mechanism of drug resistance and action in M.tb.
ANSWER: Dear Reviewer, thank you very much for your careful reading of our manuscript and for providing such valuable feedback. We have addressed all of your comments and hope that the revised version of our paper has been improved. Below, please find our detailed responses to your comments.
The revised manuscript includes all changes highlighted for easy reference. We hope that the file with tracked changes is accessible to you for review.
(x) The English could be improved to more clearly express the research.
ANSWER: The English was edited for style, grammar, and clarity throughout.
I have a few minor comments:
Line 16 in-vitro hyphenate
ANSWER: As far as I know, MDPI journals write in vitro according to their style: without hyphen and not in italic.
Otherwise, this word comes from Latin, it does not need hyphen and, usually, it should be italicized.
Line 21, in silico significant, I m not sure what that means, please rephrase
ANSWER: this was rephrased. We meant a mutation is significant based on in silico prediction (Page 1, line 21).
Line 34-37, this is a very long sentence, plz shorten
ANSWER: All this paragraph including this sentence was revised.
The entire introduction seems to be rewritten, it’s a haphazard mix as of now. The authors should t try to write a coherent story which in my view can start by saying TB needs new modalities, a few of these are … these act by … and why they chose the ones they did. This may be followed by a few lines on their methodology as to why they decided on this particular approach. This is only a suggestion alternative story lines that make the paper a worthy read may also be explored.
ANSWER: We have substantially revised and reshaped the Introduction. Several new paragraphs have been added, and other sections have been reworked to improve clarity and flow. We believe these changes have made the Introduction more logical and cohesive (Pages 2-3).
Line 267, regarding NrdH this protein has been biochemically and structurally characterised Phulera, Mande, 2013 Biochemistry.
soon after Line 267 the authors present a very useful discussion on redox related proteins, I would like to bring to the attention of authors that Mtb has a very interesting redox network relying on either small molecules like NADPH, mycothiol or proteins that help transfer electrons by utilizing Cysteines in a CXXC motif, for eg see Phulera et al Journal of the Indian Institute of Science 2014.
ANSWER: Thank you very much, we considered these very interesting studies in the revised version (Page 11, lines 341-346; new refs 56,57).
Figure 2 needs to be improved, bigger labels can be a starter, also please have it looked at some other colleagues and see if any other suggestions can make it more useful
ANSWER: Figure 2 was edited, the gene names were edited to be in high resolution.
The conclusions can really use some more work, please take a moment here to review in a little more details here what some implications of your results are
ANSWER: Conclusions were revised and partly expanded – please see the highlighted text in the ms with track changes (Page 13).
Thank you again for your comments,
Violeta Valcheva, PhD
Igor Mokrousov, PhD, DSc

Reviewer 2 Report
Comments and Suggestions for Authors
This is an interesting paper that reports on the genetic changes seen in Mycobacterium tuberculosis after exposure to novel antibacterials, aroylhydrazones and nitrofuroylamides. It will likely be an important paper for further, follow-up studies that pursue these antibacterials and their modes of action.
In the introduction, the authors should explain briefly, the origin of the aroylhydrazones and nitrofuroylamides classes of compounds.
I would like to see Table 2 reorganized to include information about each sequenced isolate. For instance, I’d like to know how many mutations were identified in each sequenced genome. Also, the numbers of sequenced strains and the mutations discussed in the text don’t seem to add up. Although it is stated ‘DNA of quality 145 and quantity sufficient for NGS was obtained from 11 samples. Genomic analysis identified 7 mutations in 4 samples.’(lines 145-146), I’d like to know what was/was not found in the other 7 samples and I’d like to have the authors identify of all of the mutations in the sequenced strains. If necessary, this information could be included in a table in the supplementary materials.
For the frameshift mutations, please indicate where in the gene this occurred – was it in the 5’ or 3’ areas of the coding region? Is there the possibility of a fairly long, truncated protein being produced? Also, for the +C insertion, is it possible that some residual activities are possible for this mutant, due to slippage of ribosome on these homopolymeric tracts?
Although not explicitly stated, the mutations that arose could reflect either changes in a drug target, or an altered physiology, allowing the bacterium to survive the effects of the drug. I’d like to see some discussion of the various mutants in the light of this consideration.
Lines 261-2: I don’t follow the logic linking the IF2 gene network to resistance – explain more clearly. Also, I think IF2 is already a known target for some antibacterials (?)
Small points:
All abbreviations should be written out in full. at first use, and probably included in a glossary/abbreviations section as well. This includes QSAR and INH, used in the introduction.
‘Complemented with’ is used twice, in Figure 1 legend and on line 291. I’m not sure what the authors men in either instance. Moreover, ‘complemented’ has a precise meaning in genetics that is likely not what the authors mean here. Re-phrase please.
Lines 215-216; The English is unclear here, as it’s unclear if they’re referring to the effects of a particular mutation in glpK, or the role of GlpK. Re-write.
In figures 1 & 2, are there not meant to be lines connecting the spheres, representing the various genes ?
Comments on the Quality of English Language
see specific comments to authors
Author Response
Answers to Reviewer 2
This is an interesting paper that reports on the genetic changes seen in Mycobacterium tuberculosis after exposure to novel antibacterials, aroylhydrazones and nitrofuroylamides. It will likely be an important paper for further, follow-up studies that pursue these antibacterials and their modes of action.
ANSWER: Dear Reviewer, thank you very much for your careful reading of our manuscript and for providing such valuable feedback. We have addressed all of your comments and hope that the revised version of our paper has been improved. Below, please find our detailed responses to your comments.
The revised manuscript includes all changes highlighted for easy reference. We hope that the file with tracked changes is accessible to you for review.
(x) The English could be improved to more clearly express the research.
ANSWER: The English was edited for style, grammar, and clarity throughout.
In the introduction, the authors should explain briefly, the origin of the aroylhydrazones and nitrofuroylamides classes of compounds.
ANSWER: The Introduction was revised and more information was provided on the tested compounds (Page 3, lines 98-118, 134-140).
I would like to see Table 2 reorganized to include information about each sequenced isolate. For instance, I’d like to know how many mutations were identified in each sequenced genome. Also, the numbers of sequenced strains and the mutations discussed in the text don’t seem to add up. Although it is stated ‘DNA of quality 145 and quantity sufficient for NGS was obtained from 11 samples. Genomic analysis identified 7 mutations in 4 samples.’(lines 145-146), I’d like to know what was/was not found in the other 7 samples and I’d like to have the authors identify of all of the mutations in the sequenced strains. If necessary, this information could be included in a table in the supplementary materials.
ANSWER: In the revised manuscript, we have carefully rechecked and reanalyzed all data, updated their presentation in the main text, modified Table 2, and created a new Table S1 to detail the mutations identified in the resistant clones.
Our primary objective was to dissect the initial response of M. tuberculosis to the tested compounds. As such, this was a short-term study involving one month of culturing. Given that M. tuberculosis is a slow-growing bacterium, it is unlikely that a large number of mutations would emerge in the colonies during this timeframe. We have documented all mutations identified in the resistant clones. However, in some clones, no mutations were detected. In these cases, the observed resistance (tolerance) to the compounds may be attributed to mechanisms unrelated to heritable genomic changes. For example, increased active efflux or upregulated transcription of certain genes could play a role, though this remains speculative and would require further investigation beyond the scope of this preliminary pilot study.
For the frameshift mutations, please indicate where in the gene this occurred – was it in the 5’ or 3’ areas of the coding region? Is there the possibility of a fairly long, truncated protein being produced? Also, for the +C insertion, is it possible that some residual activities are possible for this mutant, due to slippage of ribosome on these homopolymeric tracts?
ANSWER: We revised text and Table 2 to address this comment. The resulting proteins were greatly abridged and most likely inactivated. Besides, we cite and briefly describe the previous work on glpK that described the same mutation as conferring drug tolerance. Therefore we believe that this mutation found in our study is meaningful and also account for the tolerance of this clone to the tested aroylhydrazone (please see text lines 243-245, 280-286).
Although not explicitly stated, the mutations that arose could reflect either changes in a drug target, or an altered physiology, allowing the bacterium to survive the effects of the drug. I’d like to see some discussion of the various mutants in the light of this consideration.
ANSWER: Thank you very much for this comment. We revised the Concluding section accordingly (Page 13, 2nd paragraph). However, the mutations detected in this study mostly concern not change in the drug target but rather bacterial response to the stress conditions created by the drug. Otherwise, when we described mutations we also try to explain their meaning and provide a relevant discussion for all mutations (Page …). In revised version, we have expanded discussion on nrdH and infB mutations (lines 324-356).
Lines 261-2: I don’t follow the logic linking the IF2 gene network to resistance – explain more clearly. Also, I think IF2 is already a known target for some antibacterials (?)
ANSWER: In the revised version we added relevant references and discussion (Page 11, 1st paragraph).
Small points:
All abbreviations should be written out in full. at first use, and probably included in a glossary/abbreviations section as well. This includes QSAR and INH, used in the introduction.
ANSWER: All abbreviations were checked and spelled out on the first use. See also Page 1, a paragraph on abbreviations was added.
Complemented with’ is used twice, in Figure 1 legend and on line 291. I’m not sure what the authors men in either instance. Moreover, ‘complemented’ has a precise meaning in genetics that is likely not what the authors mean here. Re-phrase please.
ANSWER: This was revised for clarity (lines 310-33, 368-369, and legends to Fig. 1 and 2).
Lines 215-216; The English is unclear here, as it’s unclear if they’re referring to the effects of a particular mutation in glpK, or the role of GlpK. Re-write.
ANSWER: this sentence was revised (Page 9, lines 283-286).
In figures 1 & 2, are there not meant to be lines connecting the spheres, representing the various genes ?
ANSWER: This was clarified in legends. These networks were generated by the STRING online tool. Lack of connecting lines means that the genes are not interacting (to the best of the current knowledge).
Thank you again for your comments,
Violeta Valcheva, PhD
Igor Mokrousov, PhD, DSc

Reviewer 3 Report
Comments and Suggestions for Authors
Title: Genomic Insight into Primary Adaptation of Mycobacterium tuberculosis to Aroylhydrazones and Nitrofuroylamides initro
Comment on title:
Needs revision as: Genomic Insight into Primary Adaptation of Mycobacterium tuberculosis to Aroylhydrazones and Nitrofuroylamides: An invitro study
What is the main question addressed by the research?
The authors have worked on new Aroylhydrazones and Nitrofuroylamides as an invitro study to search for possible good and potent antitubercular moieties in resistant strains. They describes the resistance as multifaceted in genetic perspectives and have reached the conclusions to non-specific drug tolerance, efflux systems or counteracting oxidative stress.
2. What parts do you consider original or relevant to the field? What
specific gap in the field does the paper address?
Antibiotics and antitubercular drugs development which is a challenge for the world.
3. What does it add to the subject area compared with other published
material?
Drug resistance and antitubercular drugs.
4. What specific improvements should the authors consider regarding the
methodology? What further controls should be considered?
1. kindly add an elaboration to the details of the MIC determined in section 2.3, page 5, lines no: 121-124.
2. MIC of What? a reference, a standard or of which molecule? Kindly add a positive standard of the same section
3. Add a separate section of definition or operational definitions for key terms as highlighted in the manuscript file reviewed/attached for easy reference.
5. Are the conclusions consistent with the evidence and arguments presented?
The conclusion of abstract is not tele with the conclusion of the manuscript with in the perspective of resistance key term MDR as mention the manuscript conclusion.
Were all the main questions posed addressed? By which specific experiments?
Yes
6. Are the references appropriate?
Yes
. Any additional comments on the tables and figures and the quality of the
data.
The legends of the figures need to be revised and elaborated as advised elsewhere in the reviewed manuscript. The name of the ATCC strains shall be there as well.

Author Response
Answers to Reviewer 3
Dear Reviewer, thank you very much for your careful reading of our manuscript and for providing such valuable feedback. We have addressed all of your comments and hope that the revised version of our paper has been improved. Below, please find our detailed responses to your comments.
The revised manuscript includes all changes highlighted for easy reference. We hope that the file with tracked changes is accessible to you for review.
Comment on title:
Needs revision as: Genomic Insight into Primary Adaptation of Mycobacterium tuberculosis to Aroylhydrazones and Nitrofuroylamides: An invitro study
ANSWER: Regarding the Title and your suggested revision, this is not only in vitro study, it is also genomic and bioinformatics study. In our opinion, our original version of the title is concise and straightforward, the phrase "in vitro" is integrated smoothly into the title without breaking the flow; and it feels more streamlined. Thus, we would prefer to keep the title unchanged.
What is the main question addressed by the research?
The authors have worked on new Aroylhydrazones and Nitrofuroylamides as an invitro study to search for possible good and potent antitubercular moieties in resistant strains. They describes the resistance as multifaceted in genetic perspectives and have reached the conclusions to non-specific drug tolerance, efflux systems or counteracting oxidative stress.
What specific improvements should the authors consider regarding the methodology? What further controls should be considered?
- kindly add an elaboration to the details of the MIC determined in section 2.3, page 5, lines no: 121-124.
ANSWER: A paragraph on MIC determination was expanded (Page 4, lines 149-169).
- MIC of What? a reference, a standard or of which molecule? Kindly add a positive standard of the same section
ANSWER: Isoniazid was used as a reference drug. This information was added in the revised text (Page 4, lines 170-172).
- Add a separate section of definition or operational definitions for key terms as highlighted in the manuscript file reviewed/attached for easy reference.
ANSWER: In M&M, we added the “operational definition” of resistant clones, as suggested (Page 5, lines 185-186).
We did not add definitions of frameshift or nonsynonymous mutations because these are standard definitions in genetics, and our paper is a research article, not review nor manual.
Are the conclusions consistent with the evidence and arguments presented?
The conclusion of abstract is not tele with the conclusion of the manuscript with in the perspective of resistance key term MDR as mention the manuscript conclusion. Make it uniform with manuscript conclusion in perspective of MDR terminology.
ANSWER: I do not think we can write that these compounds may be useful to fight MDR TB (if I correctly understand this comment). It is still a lot of research before these described compounds may be considered as drugs.
We revised the Conclusions section in the main text (Page 13, lines 403-412). We did not expand the abstract: because of 200-word limit it is not possible to add general sentences about perspectives of this research in terms of MDR-TB.
Any additional comments on the tables and figures and the quality of the data.
The legends of the figures need to be revised and elaborated as advised elsewhere in the reviewed manuscript. The name of the ATCC strains shall be there as well.
ANSWER: we revised legends as you recommended in the marked-up PDF file.
Thank you again for your comments,
Violeta Valcheva, PhD
Igor Mokrousov, PhD, DSc

Round 2
Reviewer 1 Report
Comments and Suggestions for Authors
Thank you for incorporating the suggestions
Comments on the Quality of English LanguageMay need editor review